# TGFβ-Treated Placenta-Derived Mesenchymal Stem Cells Selectively Promote Anti-Adipogenesis in Thyroid-Associated Ophthalmopathy

**DOI:** 10.3390/ijms23105603

**Published:** 2022-05-17

**Authors:** Hyun-Ah Shin, Mira Park, Jasvinder Paul Banga, Helen Lew

**Affiliations:** 1Department of Ophthalmology, Bundang CHA Medical Center, CHA University, Seongnam 13496, Gyeonggi-do, Korea; sha9547@naver.com (H.-A.S.); hoohoo9979@gmail.com (M.P.); 2Faculty of Life Sciences & Medicine, King’s College London, London WC2R 2LS, UK; paul.banga@kcl.ac.uk

**Keywords:** adipogenesis, thyroid-associated ophthalmopathy, thyroid disease, TGFβ, hPMSCs, SMAD

## Abstract

Orbital fibroblasts (OFs) in thyroid-associated ophthalmopathy (TAO) are differentiated from pre-adipocytes and mature adipocytes; increased lipid and fat expansion are the major characteristics of ophthalmic manifestations. Human placental mesenchymal stem cells (hPMSCs) were reported to immunomodulate pathogenesis and suppress adipogenesis in TAO OFs. Here, we prepared transforming growth factor β (TGFβ, 20 ng/mL)-treated hPMSCs (TGFβ-hPMSCs) in order to enhance anti-adipogenic effects in vitro and in TAO mice. TAO OFs were grown in a differentiation medium and then co-cultured with hPMSCs or TGFβ-hPMSCs. TAO OFs were analyzed via quantitative real-time polymerase chain reaction, Oil red O staining, and western blotting. The results showed that TGFβ-hPMSCs reduced the expression of adipogenic, lipogenic, and fibrotic genes better than hPMSCs in TAO OFs. Moreover, the adipose area decreased more in TAO mice injected with TGFβ-hPMSCs compared to those injected with hPMSCs or a steroid. Further, TGFβ-hPMSCs inhibited inflammation as effectively as a steroid. In conclusion, TGFβ-hPMSCs suppressed adipogenesis and lipogenesis in vitro and in TAO mice, and the effects were mediated by the SMAD 2/3 pathways. Furthermore, TGFβ-hPMSCs exhibited anti-inflammatory and anti-fibrotic functions, which suggests that they could be a new and safe method to promote the anti-adipogenic function of hPMSCs to treat TAO patients.

## 1. Introduction

Thyroid-associated ophthalmopathy (TAO) frequently manifests with thyroid dysfunction and is also known as thyroid eye disease or Graves’ ophthalmopathy (GO) [1]. Thyroid eye manifestations include proptosis, eyelid retraction, exposure keratopathy, restrictive strabismus, the limitation of eye movement, compressive optic neuropathy, disfigured appearance, and increases in the orbital fat, connective tissue, and extraocular muscle volumes [2]. Orbital fibroblasts (OFs) in most TAO patients are differentiated from pre-adipocytes and mature adipocytes, and thus express high levels of lipid and fat [3]. 

Human placental mesenchymal stem cells (hPMSCs) are cells with self-renewing abilities originated from human placenta that can differentiate into multiple lineage cell types [4]. Various studies have shown that hPMSCs can produce therapeutic effects for autoimmune-, inflammation-, and hormone-related diseases via the differentiation and secretion of factors such as growth factors, cytokines, and chemokines [5,6]. We previously showed that adipogenesis was inhibited by SREBP2-HMGCR signaling when hPMSCs were cultured with TAO fibroblasts [7], which we consider to be a potential therapeutic agent for TAO. Since various orbital pathologic findings from TAO were examined, further studies about the therapeutic effects of MSCs have been expected.

Members of the transforming growth factor β (TGFβ) superfamily regulate many cellular processes, including apoptosis, inflammation, fibrosis, and adipocyte differentiation [8]. TGFβ blocks adipocyte differentiation in vitro, and the transgenic overexpression of TGFβ in adipose tissue also inhibits differentiation. TGFβ significantly reduces adipogenesis differentiation via TGFβ/SMAD signaling [9]. However, no studies have yet explored whether TGFβ stimulated hPMSCs effect on adipogenesis differentiation in TAO.

In this study, we investigated the inhibition effect on adipogenesis from functionally enhanced hPMSCs using TGFβ in TAO OFs with in vitro and in vivo tests.

## 2. Results

### 2.1. Characterization of hPMSCs Treated with TGFβ1

To investigate the effects of TGFβ1 on hPMSCs, hPMSCs were incubated with 20 ng/mL of TGFβ1 for 24 h. Cell viability decreased significantly by 25.9% (Figure 1A), and proliferation activity decreased by 39.2% in TGFβ-treated hPMSCs (Figure 1B). Next, we examined the mRNA expressions of adipogenesis, lipogenesis, and inflammation-related genes in hPMSCs treated with TGFβ1. Adipogenesis marker genes, i.e., *PPARγ*, *C/EBPα*, *LEPTIN*, and *HMGCR* expressions, were significantly decreased by TGFβ1. However, the mRNA expression of the lipogenesis marker genes *IGF-1R* and *INSIG2* were increased by TGFβ1 treatment (Figure 1C). The expression of inflammation marker genes, *TGFβ1*, *TGFβ2,* and *TNFα* mRNA levels also increased, whereas *IL6*, *IL8*, *ICAM-1,* and *IL-1β* expressions were decreased by TGFβ1 treatment (Figure 1D). These results indicated that TGFβ could strongly enhance hPMSCs in anti-adipogenesis and anti-inflammatory effects.

### 2.2. Effect of TGFβ-hPMSCs on TAO-Derived OFs

We confirmed changes in the levels of adipogenesis marker genes in the OFs of TAO patients. *PPARγ*, *C/EBPα*, *LEPTIN*, and *aP2* expression increased after the induction of adipogenesis and decreased on co-culture with hPMSCs. Reductions were greater on co-culture with TGFβ-hPMSCs than with hPMSCs. The hPMSC or TGFβ-hPMSC co-cultures were performed after the induction of OF adipogenesis by SB431542 (TGF-β RI kinase inhibitor) (Figure 2A). The extent of the differentiation of normal and TAO OFs was examined via Oil Red O staining. Both adipogenesis-induced normal and TAO OFs exhibited significantly more lipid accumulation compared to the negative control. After co-culture with hPMSCs and TGFβ-hPMSCs, lipid accumulation in TAO OFs was inhibited by 65.8% and 70.8%, respectively, but not when the TAO OFs were pre-treated with SB431542 (Figure 2B). TAO OFs co-cultured with hPMSCs also regulated the mRNA expressions of lipogenic genes, including *INSIG1, INSIG2, SREBP2, HMGCR,* and *IGFBP3* (Figure 2C). In adipogenic TAO OFs, hPMSCs and TGFβ-hPMSCs increased the levels of the mRNA expression of *INSIG1*, and hPMSCs was more effective than TGFβ-hPMSCs. *INSIG1* and *INSIG2* play roles in cholesterol homeostasis and precursor adipocyte differentiation [10]. Thus, both hPMSCs and TGFβ-hPMSCs regulated adipocyte differentiation. In adipogenic TAO OFs, the levels of the mRNA of *SREBP2*, *HMGCR,* and *IGFBP3* significantly increased compared to those of non-adipogenic TAO OFs. TGFβ-hPMSCs inhibited the expressions of these genes more so than the naïve hPMSCs did (Figure 2C). To investigate whether hPMSCs were involved in other reactions, we confirmed the mRNA expressions of pro-fibrotic genes including *α-SMA* and *FIBRONECTIN*, as well as a gene related to inflammation, *IL-17*. The expression levels increased in adipogenic TAO OFs, and co-culture with TGFβ-hPMSCs inhibited expression more effectively than did co-culture with naïve hPMSCs. We used SB431542 to confirm the TGFβ-mediated anti-adipogenic effects of TGFβ-hPMSCs (Figure 2D). Thus, hPMSCs inhibited adipogenesis, lipogenesis, fibrosis, and inflammation in adipogenic TAO OFs, and the inhibitory effects of TGFβ-hPMSCs were greater than those of hPMSCs.

### 2.3. Pathology Assessment 

We assessed the pathologies of TAO groups and analyzed adipogenesis and inflammation-related target protein expressions in orbital tissues of TAO animals. We measured the adipose area around the optic nerve. The average of the TAO animals (Sham) was significantly larger (206.5%) than the negative control animals (NC). The injection of hPMSCs or a steroid (STE) significantly decreased the adipose area by 72.9% and 72.3%, respectively. Particularly, TGFβ-hPMSC injection significantly inhibited the adipose area by 79.6% (Figure 3A).

### 2.4. TGFβ-hPMSCs Inhibit Adipogenesis and Inflammation in TAO Animals

We next examined the levels of the adipogenic proteins Igf-1r, Pparγ, and C/ebpα. Pparγ levels were significantly reduced by hPMSCs, and TGFβ-hPMSCs reduced the Igf-1r, Pparγ, and C/ebpα levels in the TAO mice (Figure 3B). We then examined the levels of the inflammatory proteins Icam-1, Tgfβ1, and Il6 and the Tnfα protein. TGFβ-hPMSCs more effectively inhibited Icam-1 and Tgfβ by 57.9% and 57.4%, respectively, compared to 45.2% and 45.1% by hPMSCs (Figure 3C). 

### 2.5. TGFβ-hPMSCs Regulate TAO OFs via the SMAD Pathway

To investigate the associated pathway regarding the inhibition of adipogenesis and lipogenesis, we examined the protein expression of SMAD family. We analyzed the phosphorylation of SMAD2 (p-SMAD2) and SMAD3 (p-SMAD3) in TAO OFs. Compared to the negative control, p-SMAD2 and p-SMAD3 levels decreased in adipogenic TAO OFs and hPMSCs co-cultured TAO OFs. However, TGFβ-hPMSCs increased p-SMAD3 by 106.1% and significantly increased p-SMAD2 by 137.8% more than adipogenic TAO OFs (Figure 4A). Additionally, we analyzed the p-Smad2 and p-Smad3 protein levels in the TAO mice (Figure 4B). The p-Smad3 level was not regulated by treatments in TAO mice. Human PMSCs significantly reduced p-Smad2 more than the sham group, and TGFβ-hPMSCs significantly increased p-Smad2 expression by 161.3% (Figure 4B). From these results, we could assume that TGFβ-hPMSCs inhibited the activities of adipogenic, lipogenic, and fibrotic genes through the SMAD2/3 pathway (Figure 5).

## 3. Discussion

Mesenchymal stem cells (MSCs), a kind of cell with self-renewal and multiple differentiation characteristics, are known for broad applications in the field of regenerative medicine [11,12,13]. A series of MSCs have been isolated from different tissue sources, including bone marrow, adipose, and placenta. We recently showed that human placenta-derived MSCs (hPMSCs) inhibited adipogenesis in TAO OFs [7]. However, adipogenesis is the process of differentiating pre-adipocytes into mature adipocytes, in which the transcriptional activators C/EBPα and PPARγ genes are both necessary. These transcription factors are activated, leading to growth arrest for terminal differentiation and the expression of adipocyte genes, including FABP4/aP2 and LEPTIN [14]. Regarding the gene expressions of adipogenesis, hPMSCs mostly inhibited them in TAO OFs. The expressions of PPARγ, C/EBPα, and aP2 were not significantly inhibited, but that of INSIG2 was increased in adipogenic TAO OFs co-cultured with hPMSCs. Thus, we applied TGFβ-treated hPMSCs to enhance the anti-adipogenesis of hPMSCs in TAO.

TGFβ profoundly affects the differentiation of many cell types of mesenchymal origin, including pre-adipocytes, osteoblasts, and myoblasts [15]. Members of the TGFβ family regulate MSC lineage selection and progression of mesenchymal differentiation to specific cell types by controlling the expression and activity of key transcription factors [16]. TGFβ family members, including BMPs, TGFβs, activins, and inhibins, regulate the differentiation of early bone marrow stromal cells (BMSCs) into the mature matrix-secreting osteoblasts and osteocytes required for normal late myoblast differentiation, potentially via Smad-independent mechanisms, which are a sub-signal of TGFβ [17]. In addition, many studies have reported that inflammation and adipogenesis are regulated by TGFβ, which inhibits adipogenesis in unloaded bone marrow stromal cells [18] and also inhibits adipocyte differentiation by SMAD3 interacting with the CCAAT/Enhancer-binding Protein (C/EBP) [19]. We found that TGFβ enhanced the anti-adipogenic effects of hPMSCs on TAO both in vitro and in vivo. TGFβ-hPMSCs decreased *IGFBP3* expression, indicating that adipogenesis was inhibited via signals other than IGFBPs. The powerful anti-adipognesis effect of TGFβ-hPMSCs was mediated via SMAD signaling.

The SMAD family of transcription factors interacts with TGFβ receptors to propagate signals. The receptor’s two transmembrane protein serine/threonine kinases, receptor types I and II, are brought together by the ligand, which acts as a receptor assembly factor. Type I receptors specifically recognize receptor-activated SMADs (R-SMADs); these include SMAD2/3, which is also recognized by the TGFβ and activin receptors, and SMAD1/5/8, which is recognized by BMP receptors [20]. TGFβ binding activates SMAD2/3 and suppresses pre-adipocyte differentiation. However, BMP-like ligands, which primarily activate the SMAD1/5/8 pathways, increase adipocyte numbers [21]. Activated SMAD2/3 inhibited adipogenesis and this inhibition could be blocked by the addition of the SB431542 (inhibitor of SMAD2/3) [22,23,24].

Furthermore, latent TGFβ1 is known to protect from renal fibrosis and inflammation [22,25] and to suppress T-cell proliferation and activation through Treg differentiation [26]. S. Sana et al. proposed a relationship between SMAD proteins and TGFβ signaling, in which high SMAD7 immunoreactivity and a lack of p-SMAD3 expression caused defective TGFβ signaling in environmental enteropathy [27]; this was similar to a previous result in inflammatory bowel disease and Crohn’s disease [28]. We found that the anti-inflammatory effects of TGFβ-hPMSCs in vivo were as good as those of a steroid and better than those of hPMSCs. Thus, TGFβ-hPMSCs may be used to treat both inflammatory diseases and TAO. In this study, hPMSCs alone inhibited adipogenesis, but TGFβ enhanced such inhibition. We previously showed that hPMSCs have immune modulatory effects, inhibiting adipogenesis via anti-inflammatory effects [29]. TGFβ induced SMAD2/3 signaling in anti-inflammation conditions; therefore, it could provide an adipogenesis inhibitory function. The phosphorylation of TGFβ induced a SMAD2/3 decrease in TAO OFs treated with the TGFβ inhibitor SB431542. 

In addition, TGFβ plays a key role in fibrosis [30,31]. Fibroblasts are activated by TGFβ to transition into myofibroblasts, a key effector cell. Myofibroblasts are characterized by high levels of contractile proteins, including α-smooth muscle actin (α-SMA), collagens I, III, V, and fibronectin [32,33]. However, we found that when hPMSCs treated with TGFβ were co-cultured with TAO OFs, the α-SMA and fibronectin levels fell. We expected that TAO OF fibrosis would be aggravated on co-culture with TGFβ-hPMSCs, because adipogenesis was induced. However, we observed the opposite. CD4+ T helper cells are involved in renal inflammation and fibrosis, and it has been reported that hPMSCs can convert an inflammatory environment into an anti-inflammatory environment by affecting the polarization of CD4+ T cells and macrophages [34]. Therefore, even when fibrosis is stimulated, TAO OFs are expected to suppress fibrosis due to the influence of hPMSCs, rather than that of TGFβ. TGFβ-hPMSCs did not stimulate the fibrosis of TAO OFs, but rather inhibited differentiation into myoblasts as well as adipogenesis. Further research on the role played by TGFβ-hPMSCs in TAO fibrosis is needed. 

In conclusion, we found that TGFβ-hPMSCs selectively enhanced the anti-adipogenic effects of TAO compared with naïve hPMSCs both in vitro and in vivo. These results show that TGFβ-hPMSCs activated SMAD2/3, and phosphorylated SMAD2/3 suppressed the transcription of lipogenic genes (e.g., HMGCR and SREBP2) and adipogenic genes (e.g., PPARγ, C/EBPα, Leptin, and aP2), as well as fibrotic genes, including a-SMA and Fibronectin. This study provides a new and safe method to enhance the anti-adipogenic function of hPMSCs that can be used to treat TAO patients.

## 4. Material and Methods

### 4.1. Cell Preparation of hPMSCs and Orbital Fibroblast

The orbital fibroblast isolation protocol was approved by the Institutional Review Committee (IRB-2018-01-007) of Bundang Cha Hospital in Seongnam, Korea, and all patients’ consent was obtained. Ophthalmic adipose tissue descriptions were obtained from patients with TAO during intraocular adipose decompression and from controlled individuals without a history of TAO during orbital plastic surgery. The tissue was chopped and treated with collagenase (0.25 mg/mL; Thermo Fisher Scientific, Waltham, MA, USA) at 37 °C for 1 h in a shaking incubator. After incubation, the digested tissues were placed directly in culture dishes with DMEM/F12 containing 20% fetal bovine serum (FBS; Thermo Fisher Scientific) and 1% penicillin/streptomycin (Thermo Fisher Scientific). Experiments were performed using cells from the fifth to the eighth passage. Human placenta stem cell preparation and culturing were conducted as previously reported [35]. Human PMSCs were cultured in α-modified minimal essential medium (α-MEM; HyClone, Logan, UT, USA) supplemented with 10% FBS (Thermo Fisher Scientific), 1% P/S (Thermo Fisher Scientific), 1 μg/mL heparin (Sigma-Aldrich, St. Louis, MO, USA), and 25 ng/mL human fibroblast growth factor-4 (hFGF-4; Peprotech, Rocky Hill, NJ, USA). 

### 4.2. Adipocyte Differentiation 

Using a six-well plate, normal and TAO-derived OFs (4 × 10^4^/cm^2^) were seeded per well and incubated in DMEM medium supplemented with 33 μM biotin, 17 μM pantothenic acid, 0.2 nM triiodothyronine (T_3_), 10 μg/mL transferrin, 0.2 μM carbaprostacyclin (cPGI_2_; Cayman Chemical, Ann Arbor, MI, USA), 0.1 mM isobutylmethylxanthine (IBMX), 1 μM dexamethasone, and 1 μM insulin (all from Sigma-Aldrich). During the initial differentiation stage, the incubation medium was replaced daily for 4 days. For the maturation of adipocytes, the cells were incubated in a differentiation medium without 1 μM dexamethasone, 0.1 mM IBMX and 1 μM insulin (all from Sigma-Aldrich). Cells were harvested at 2, 4, and 10 days for experiment.

### 4.3. Co-Culture Experiments

Before co-culturing, hPMSCs were treated with recombinant human TGFβ1 at 20 ng/mL (Peprotech) for 24 h, and then adipose-induced normal and TAO-derived OFs were co-cultured with naïve hPMSCs (OF + hPMSCs) or TGFβ-hPMSCs (OF+ TGFβ-hPMSCs) using Transwell inserts (8 μm pore size; Corning, NY, USA) for 24 h at 37 ℃ in a humidified atmosphere containing 5% CO_2_. In addition, OFs were pretreated with 10μM of TGFβ receptor kinase inhibitor, SB431542 (Selleckchem, Houston, TX, USA), just before co-culture with hPMSCs (SB + OF + hPMSCs) or TGFβ-hPMSCs (SB + OF + TGFβ-hPMSCs). Cells were harvested after 24 h.

### 4.4. CCK-8 Assay

The hPMSCs were seeded at a density of 1 × 10^4^ cells/well in 96 well plates. After TGFβ treatment, the cells were incubated with 10 μL of Cell Counting Kit-8 solution (CCK-8; Dojindo, Kumamoto, Japan) for 2 h at 37 °C. Then, a microplate reader (SpectraMax iD5, Molecular Devices, San Jose, CA, USA) was used to determine the absorbance at 450 nm.

### 4.5. BrdU Incorporation Assay

The hPMSCs were seeded at a density of 1 × 10^4^ cells/well in 96 well plates. After TGFβ treatment, cell proliferation was examined using the BrdU cell proliferation ELISA kit (Cell Biolabs, INC., San Diego, CA, USA) according to the manufacturer’s instructions. The absorbance value was measured using a microplate reader (SpectraMax iD5) at 450 nm.

### 4.6. Oil Red O Staining

Oil Red O staining was performed on the tenth day of differentiation. Cells were washed twice with phosphate-buffered saline (PBS), fixed with 10% formalin for 30 min, stained with Oil-Red O for 1 h, and washed twice with water. After Oil Red O staining, the cells were dissolved in isopropanol and quantified by measuring absorbance at 470 nm, and were then photographed by a phase-contrast microscope (Olympus CKX41, Tokyo, Japan) at 20× magnification.

### 4.7. Quantitative Real-Time Polymerase Chain Reaction

Using TRIzol reagent (Ambion, Carlsbad, CA, USA), we isolated RNA from human OFs for cDNA synthesis. According to the manufacturer’s protocol, we synthesized cDNA using 1 ug of RNA. Gene expression was quantified with amfiSure qGreen Q-PCR Master Mix, Low ROX (GenDEPOT, Katy, TX, USA) and calculated by the delta delta CT method, and real-time PCR reactions were performed using a QuantStudio^TM^1 Real-Time PCR Instrument (Applied Biosystems, Foster City, CA, USA). The sequences of the primers used are presented in Table 1.

### 4.8. Development of an Experimental Mouse Model of TAO Using Female BALB/c Mice

The pTriEx1.1Neo-hTSHRA A-subunit Plasmid was helpfully provided by Professor Banga [36]. We generated a TAO disease model by the injection of pTriEx1.1Neo-hTSHR A-subunit plasmid to the leg muscle using electroporation and the characterized TAO disease model in our previous report [29]. The animals undergoing experimental TAO were divided into four groups injected in the orbit (intra orbital injection): a treatment group injected with hPMSCs (3 × 10^5^ cells/30 µL), a treatment group injected with TGFβ-hPMSCs (3 × 10^5^ cells/30 µL), a treatment group injected with steroids (0.4 mg/each, triamcinolone acetonide, Dongkwang Pharmaceutical Co., Hanmi, South Korea), and a sham group (30 µL BSS PLUS). Intra-orbital injection was performed on the left orbit. One week after hPMSCs injection, the animals were sacrificed, after which orbital tissue was excised for histopathological analyses. 

### 4.9. Orbital Tissue Histopathology

To quantify the adipose area around the optic nerve, histopathology was conducted using a ZEISS Axio Scan Z1 slide scanner (Carl Zeiss, Jena, Germany). The cross-sectional area of the orbital fat was normalized to the reverse-side adipose tissue area of each mouse. The adipose areas of the orbital sections of each mouse were evaluated in every group.

### 4.10. Western Blot Analysis

Protein lysates were prepared from animal orbital tissues and human OFs; orbital tissues from each group were homogenized with PRO-PREP solution (Intron, Gyeonggido, Korea). A total of 20 ug proteins were separated by sodium dodecylsulfate (SDS)-polyacrylamide gel electrophoresis and transferred to the membrane. Membranes transferred from gels were incubated with primary antibodies such as anti-PPARγ (GeneTex, Irvine, CA, USA), C/EBPα (GeneTex), IGF-1R (GeneTex), p-SMAD3 (GeneTex), p-SMAD2 (GeneTex), IL-6 (GeneTex), TNFα (GeneTex), ICAM-1 (Thermo Fisher Scientific), TGFβ1 (GeneTex), and β-actin (Santa Cruz Biotechnology, Inc., Dallas, TX, USA). After washing, second antibodies (GeneTex), horseradish peroxidase-conjugated anti-rabbit or mouse IgG were diluted 1:5000 and were incubated with membranes at room temperature for 2 h. The target protein bands were detected with enhanced chemiluminescence (ECL) solution (Bio-Rad Laboratories, Hercules, CA, USA) using an ImageQuant LAS 4000 (GE Healthcare Life Sciences, Little Chalfont, UK). 

### 4.11. Statistical Analyses

Data analyses were performed using GraphPad Prism (GraphPad Software, La Jolla, CA, USA). Significant differences were identified using a *t*-test or nonparametric statistical test, followed by a Mann–Whitney U-test at a 5% significance level. 

## Figures and Tables

**Figure 1 ijms-23-05603-f001:**
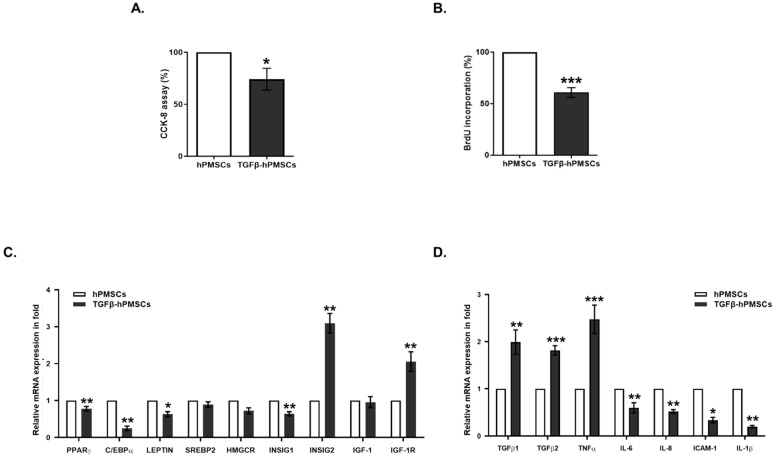
Characterization of hPMSCs with TGFβ treatment. (**A**) Cell viability was determined by CCK-8 assay and (**B**) BrdU assay at 24 h after TGFβ (20 ng/mL) treatment. (**C**) The relative mRNA expressions of adipogenesis and lipogenesis marker genes (e.g., *PPARγ*, *C/EBPα*, *LEPTIN*, *SREBP2*, *HMGCR*, *INSIG1*, *INSIG2*, *IGF-1*, and *IGF-1R*) were analyzed by RT-qPCR after TGFβ treatment. (**D**) The relative mRNA expressions of inflammation marker genes (e.g., *TGFβ1*, *TGFβ2*, *TNFα*, *IL-6*, *IL-8*, *ICAM-1*, and *IL-1β*) were analyzed by RT-qPCR at 24 h after TGFβ treatment (* *p* < 0.05, ** *p* < 0.01, *** *p* < 0.001 vs. hPMSCs).

**Figure 2 ijms-23-05603-f002:**
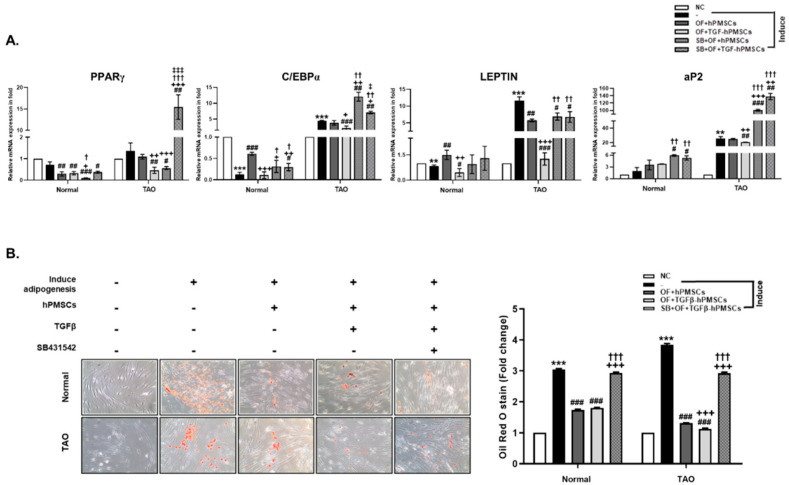
TGFβ-hPMSCs inhibit adipogenesis, lipogenesis, and fibrosis in TAO-derived OFs. (**A**) The mRNA expressions of adipogenesis marker genes (e.g., *PPARγ*, *C/EBPα*, *LEPTIN*, and *aP2*) were analyzed by RT-qPCR. (**B**) Quantification of ORO staining in differentiated OFs. The graph shows relative quantification at 470 nm absorbance. Data were presented as a fold change (mean± SEM). (**C**) The mRNA expressions of lipogenesis genes (e.g., *IGFBP3*, *INSIG1*, *INSIG2*, *SREBP2*, and *HMGCR*). (**D**) Fibrosis and inflammation genes (e.g., *α-SMA*, *FIBRONECTIN*, and *IL17*) were analyzed by RT-qPCR. Significantly different values between the groups are indicated with marks (* *p* < 0.05, ** *p* < 0.01, *** *p* < 0.001 Negative control vs. adipogenesis-induced OFs (-); # *p*  <  0.05, ## *p* < 0.01, ### *p* < 0.001 vs. adipogenesis-induced OFs (-); + *p* < 0.05, ++ *p* < 0.01, +++ *p*< 0.001 vs. OF + hPMSCs; † *p* < 0.05, †† *p* < 0.01, ††† *p* < 0.001 vs. OF + TGFβ-hPMSCs; ^‡^
*p* < 0.05, ^‡‡‡^
*p* < 0.001 vs. SB + OF + hPMSCs). The time-dependent manner of gene expressions was analyzed as follows: day 2 (*PPARγ*); day 4 (*C/EBPα*, *LEPTIN*, *SREBP2*, *HMGCR*, and *IL-17*); day 10 (*aP2*, *IGFBP3*, *INSIG1*, *INSIG2*, *α-SMA*, and *FIBRONECTIN*).

**Figure 3 ijms-23-05603-f003:**
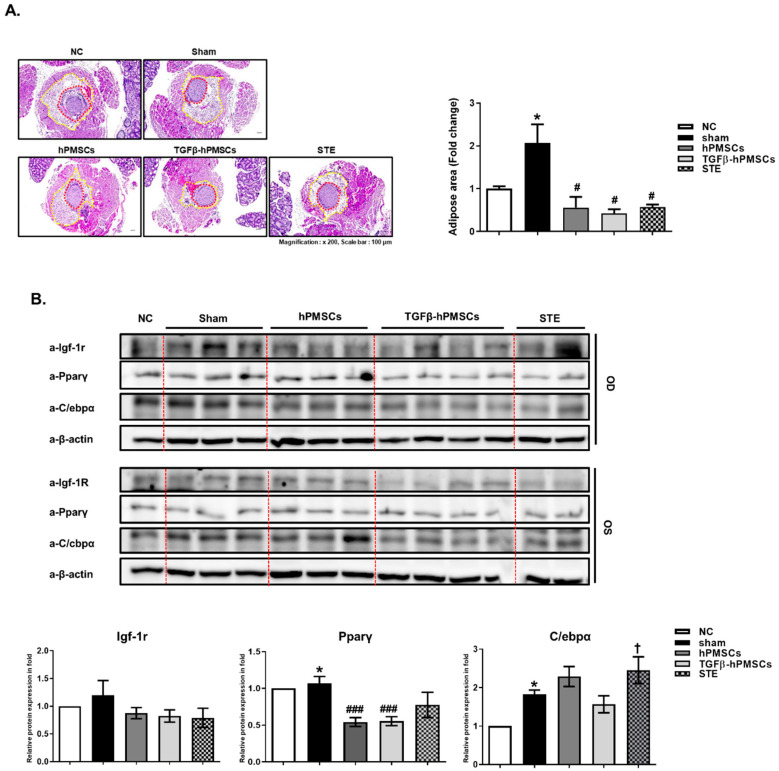
Histologic analysis of TAO animals treated with hPMSCs, TGFβ-hPMSCs, and steroid injection. (**A**) H and E-stained section from TAO mice with an expansion of adipose area around the optic nerve (magnification 100×). Data were presented as a fold change (mean ± SEM) of adipose area volume around optic nerve compared with the sham (* *p* < 0.05 Negative control vs. Sham; # *p* < 0.05 vs. Sham). Negative control *n* = 3, Sham *n* = 4, hPMSCs *n* = 3, TGFβ-hPMSCs *n* = 3, STE (steroid) *n* = 3. By western blotting, (**B**) adipogenesis-related protein (e.g. Igf-1r, Pparγ and C/ebpα) expressions of orbital tissues were analyzed. Sham *n* = 3, hPMSCs *n* = 3, TGFβ-hPMSCs *n* = 4, STE (steroid) *n* = 2. (**C**) Inflammation-related protein (e.g., Icam-1, Tgfβ1, Il-6, and Tnfα) expressions of orbital tissues were analyzed. Sham *n* = 3, hPMSCs *n* = 3, TGFβ-hPMSCs *n* = 2, STE (steroid) *n* = 2. Data were presented as a fold change (mean± SEM). Expression levels were normalized to β-actin, and the values of OS were divided OD (* *p* < 0.05 Negative control vs. Sham; # *p* < 0.05, ## *p* < 0.01, ### *p* < 0.001 vs. Sham; † *p* < 0.05 vs. TGFβ-hPMSCs). OD, oculus dexter; OS, oculus sinister.

**Figure 4 ijms-23-05603-f004:**
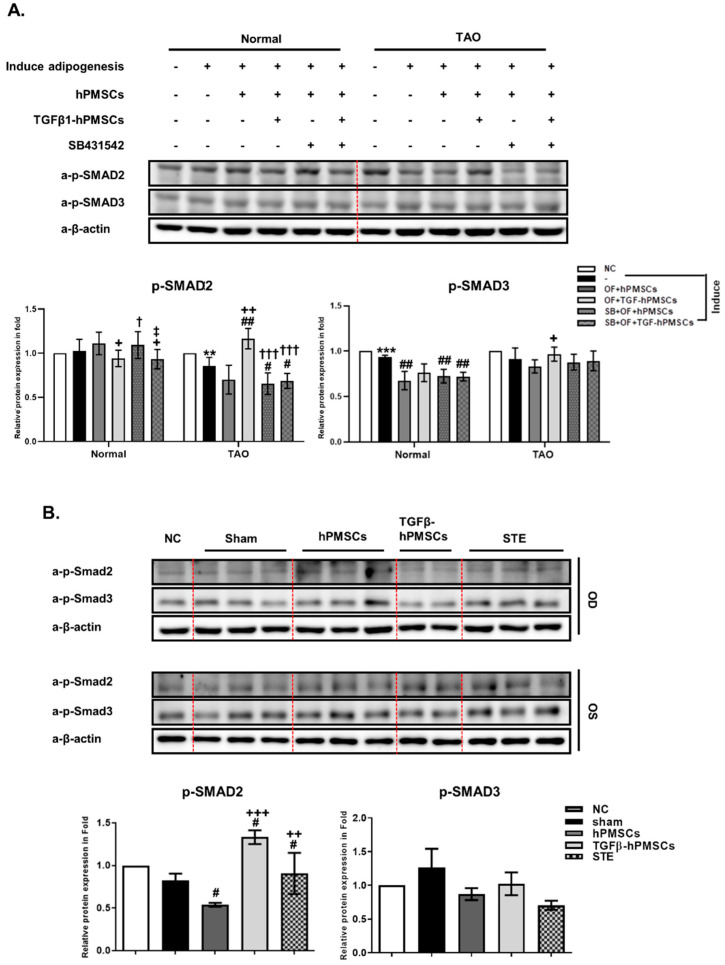
TGFβ-hPMSC co-culture regulated SMAD signaling pathway. (**A**) Protein lysates from TAO OFs were used to western blot for analysis of phosphorylation-SMAD protein on day 10 (e.g., p-SMAD2 and p-SMAD3) expression levels (** *p* < 0.01, *** *p* < 0.001 vs. Negative control vs. adipogenesis induced OFs; # *p* < 0.05, ## *p* < 0.01 vs. adipogenesis-induced OFs; + *p* < 0.05, ++ *p*<0.01 vs. OF + hPMSCs; † *p* < 0.05, ††† *p* < 0.001 vs. OF + TGFβ-hPMSCs; ^‡^
*p* < 0.05 vs. SB + OF+ hPMSCs). (**B**) Mouse orbital tissue lysates were investigated by western blot to detect *p*-Smad2 and p-Smad3 expression levels. Expression levels were normalized to β-actin, and the values of OS were divided OD. Data were presented as a fold change (mean± SEM) (# *p* < 0.05 vs. Sham; ++ *p* < 0.01, +++ *p* < 0.001 vs. hPMSCs). Sham *n* = 3, hPMSCs *n* = 3, TGFβ-hPMSCs *n* = 2, STE (steroid) *n* = 3.

**Figure 5 ijms-23-05603-f005:**
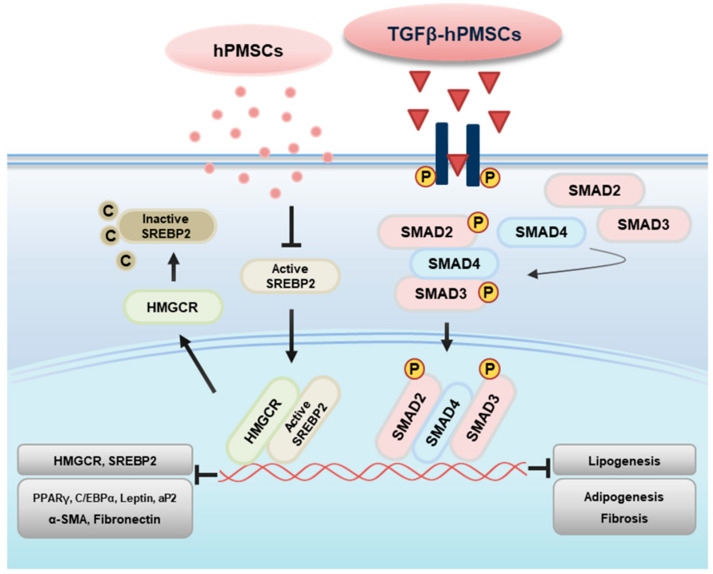
Proposed pathway of TGFβ-hPMSCs regarding anti-adipogenesis in TAO. TGFβ-hPMSCs stimulate TGFβ signal transduction by binding to transmembrane receptors. The phosphorylated receptors transmit signaling by phosphorylating Smad 2/3 and inducing their binding to Smad 4. The Smad complex translocates into the nucleus and acts as transcription factors to inhibit lipogenic gene (e.g., HMGCR and SREBP2) and adipogenic gene (e.g., PPARγ, C/EBPα, Leptin, and aP2) transcription, as well as fibrotic gene transcription, including a-SMA and Fibronectin.

**Table 1 ijms-23-05603-t001:** Human primer sequences using quantitative real-time polymerase chain reaction.

Genes		Primer Sequences	Tm
PPARγ	Forward	5’-TTGACCCAGAAAGCGATTCC-3’	50
	Reverse	5’-AAAGTTGGTGGGCCAGAATG-3’	
C/EBPα	Forward	5’-TGTATACCCCTGGTGGGAGA -3’	50
	Reverse	5’-TCATAACTCCGGTCCCTCTG-3’	
LEPTIN	Forward	5’-GGTTGCAAGGCCCAAGAA-3’	50
	Reverse	5’-ACATAGAAAAGATAGGGCCAGC-3’	
AP2	Forward	5’-ATGGGGGTGTCCTGGTACAT-3’	60
	Reverse	5’-ACGTCCCTTGGCTTATGCTC-3’	
SREBP2	Forward	5’-AACGGTCATTCACCCAGGTC-3’	60
	Reverse	5’-GGCTGAAGAATAGGAGTTGCC-3’	
HMGCR	Forward	5’-AAGGAGGCATTTGACAGCAC-3’	60
	Reverse	5’-CTGACCTGGACTGGAAACG-3’	
IGFBP3	Forward	5’-AAGTTGACTACGAGTCTCAG-3	60
	Reverse	5’-ACGGCAGGGACCATATTC-3	
α-SMA	Forward	5’-CTCCCAGGGCTGTTTTCCCA-3’	60
	Reverse	5’-CCATGTCGTCCCAGTTGGTG-3’	
FIBRONECTIN	Forward	5’-CCAAGAAGGGCTCGTGTG-3’	60
	Reverse	5’-TGGCTGGAACGGCATCA-3’	
IL-17	Forward	5’-CTGTCCCCATCCAGCAAGAG-3’	60
	Reverse	5’-AGGCCACATGGTGGACAATC-3’	
INSIG1	Forward	5’-GGCAGCTTCCCAAGTATTCG-3’	55
	Reverse	5’-AGCACCATCAACCTACCTCCT-3’	
INSIG2	Forward	5’-TCACACTGGCTGCACTATCC-3’	60
	Reverse	5’-ACAGTTGCCAAGAAGGCAAT-3’	
IGF-1	Forward	5’-CATGTCCTCCTCGCATCTCT-3’	50
	Reverse	5’-GGTGCGCAATACATCTCGAG-3’	
IGF-1R	Forward	5’-AGAAGGAGGAGGCTGAATAC-3’	55
	Reverse	5’-GGTCGGTGATGTTGTAGGT-3’	
TGFβ1	Forward	5’-CTGGACACCAACTATTGC-3’	50
	Reverse	5’-CTTCCACCCGAGGTCCTT-3’	
TGFβ2	Forward	5’-ATTGCCCTCCTACAGACTTGAG-3’	60
	Reverse	5’-CAGCACAGAAGTTGGCCATTGTA-3’	
TNFα	Forward	5’-GGTGATCGGTGCCAACAAGGA-3’	60
	Reverse	5’-CACGCTGGCTCAGCCACTG-3’	
IL-6	Forward	5’-TGAGAAAGGAGACATGTAACAAGAGT-3’	60
	Reverse	5’-TTGTTCCTCACTACTCTCAAATCTGT-3’	
IL-8	Forward	5’-ACTCCAAACCTTTCCACC-3’	60
	Reverse	5’-CTTCTCCACAACCCTCTG-3’	
ICAM-1	Forward	5’-CAGTCACCTATGGCAACGACT-3’	50
	Reverse	5’-CTCTGGCTTCGTCAGAATCAC-3’	
IL-1β	Forward	5’-ATGAGTGCTCCTTCCAGGA-3’	60
	Reverse	5’-GATAGGTTCTTCAAAGATG-3’	
GAPDH	Forward	5’-TCCTTCTGCATCCTGTCAGCA-3’	60
	Reverse	5’-CAGGAGATGGCCACTGCCGCA-3’	

## Data Availability

The data that support the findings of this study are available from the corresponding author upon reasonable request.

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
