# Peer review of "TGFβ-Treated Placenta-Derived Mesenchymal Stem Cells Selectively Promote Anti-Adipogenesis in Thyroid-Associated Ophthalmopathy"

_ijms, 2022, doi:10.3390/ijms23105603_

Round 1

Reviewer 1 Report

In this study, Shin et al studied the effects of TGFβ pre-treated PhMSCs (TGFβ-hPMSCs ) on TAO-derived orbital fibroblasts (TAO OFs) and TAO mouse models. They found that TGFβ-hPMSCs reduced the expression of adipogenic, lipogenic, and fibrotic genes greater than hPMSCs in TAO OFs, and reduced the adipose area of TAO mice. Overall, this is an interesting finding, and TGFβ-hPMSCs may become a novel option for TAO patients. I have several concerns, however.

  1. The mechanism of this effect is still unclear. The TGFβ pre-treatment is only 24 hours. The gene expression profile changes of adipogenesis, lipogenesis, inflammation, and fibrosis of hPMSCs cannot be explained by this short treatment. Especially the effects on fibrosis cannot be explained. The authors may need to do RNA-sequencing to get an overview of the transcriptome changes of hPMSCs before and after TGFβ treatment.
  2. Only one concentration of TGFβ1 (20ng/ml) was tested. Did the authors test other concentrations or duration (2-3 days)?
  3. If add TGFβ directly to the co-culture system of hPMSCs/ TAO OFs, do they have a similar effect?
  4. There are many English typos.

Author Response

Answers to the reviewers’ comments

I read all the comments and rewrote the phrases according to your suggestions.

And I also tried to explain the ideas and answered to every question from you.

I really appreciate your professional comments and excellent points to improve my manuscript for the readers.

Reviewer 1

Major points:

  1. The mechanism of this effect is still unclear. The TGFβ pre-treatment is only 24 hours. The gene expression profile changes of adipogenesis, lipogenesis, inflammation, and fibrosis of hPMSCs cannot be explained by this short treatment. Especially the effects on fibrosis cannot be explained. The authors may need to do RNA-sequencing to get an overview of the transcriptome changes of hPMSCs before and after TGFβ treatment.

->First, we deeply agree with your point regarding the mode of action hPMSCs after TGFβ treatment. It was also the cause for us to investigate the functional enhancement effect on hPMSCs using TGFβ pre-treatment in Thyroid associated ophthalmopathy (TAO). Instead of microarray for the transcriptome changes of hPMSCs before and after TGFβ treatment, we did our best to analyze the protein level of each gene from adipogenesis, lipogenesis, inflammation, and fibrosis detected by PCR due to the limit of time.

We sincerely hope that this result could help you review the enhancement effect of renowned cytokine TGFβ on hPMSCs and consider it favorable as a new treatment method in TAO.

=> As you recommended, we performed the western analysis to check whether the corresponding protein level was correlated with the genetic expression of adipogenesis (PPARγ, C/EBPα, aP2), lipogenesis (HMGCR, INSIG1, IGF-1R), inflammation (TGFβ1, TNFα, IL-6, ICAM-1), and fibrosis (α-SMA, IL-17) detected by RT-qPCR.

-> As a result, INSIG1, IGF-1R, TGFβ1, TNFα and ICAM-1 increased or decreased parallel to the RT-qPCR result (Fig. 1C and 1D), but PPARγ, C/EBPα, HMGCR and IL-6 did not.

TGFβ treated hPMSCs did not present the significant change of the corresponding protein level of fibrotic genes confirmed by RT-qPCR, such as α-SMA and IL-17.

We assumed that the reason why the levels of mRNA of each gene were not in accordance with those of the protein was that regulation time for each gene would be different. However, even though the change of the expression in the proteins or mRNAs were limited in TGFβ treated hPMSCs, they could modify the inflammation process of TAO OFs in terms of adipogenesis, inflammation and fibrosis in vitro and in vivo studies through the immune modulatory potential of hPMSCs in the TAO niche.    

(We ask your generosity for incomplete panel of all genes from the list. Since aP2 primer and fibronectin antibodies were out of stock, only the levels of the protein or mRNA were presented for the limit of time to prepare.)

  1. Only one concentration of TGFβ1 (20ng/ml) was tested. Did the authors test other concentrations or duration (2-3 days)?

-> We think it is a reasonable question for the concentration and duration of TGFβ treatment with hPMSCs. The concentration was decided based on the previous dose effect study of TGFβ on hPMSCs regarding the genetic expression of pro- and anti-lipogenesis (Insig-1 & 2) and anti- (IL-8) and pro-inflammation (ICAM-1, IL-1β).

When it comes to fibrosis, TGFβ 20ng/mL was the most favorable concentration in the three fibrotic genes (α-SMA, Fibronectin, and IL-17). Furthermore, TGFβ 20ng/mL turned out effective in terms of decrease of lipogenic and inflammation genes and increase of anti-lipogenic and anti-inflammatory genes in vitro study.

-> As for the of TGFβ treatment time, the duration was selected according to the previous studies.

Reference) Runguang, L.; Liang, L.; Yonggang, D.; Zeping, H.; Huiting, M.; Yaning, W.; Bin, Y., Mechanical stretch inhibits mesenchymal stem cell adipogenic differentiation through TGFβ1/Smad2 signaling. Journal of Biomechanics 2015, 48, 3665-71.

->In this paper, MSCs were cultured in adipogen ic medium with or without TGFβ1,4 ng/ml for 24h incubation, and then mechanical stretch was applied daily for 5 days. After that, the level of adipogenic differentiation gene was confirmed.

Reference) Chieh-Chih, T.; Shi-Bei, W.; Hui-Chuan, K.; Yau-Huei, W., Essential role of connective tissue growth factor (CTGF) in transforming growth factor-β1(TGF-β1)-induced myofibroblast transdifferentiation from Graves’orbital fibroblasts. Scientific Reports 2018, 8, 7276.

-> TGFβ1 was treated on orbital fibroblasts from GO patients under conditions of 5, 10, 20, 40 ng/ml, and 24 h incubation, and changes in fibrosis-related genes were confirmed (CTGF, Fibronectin and α-SMA). As a result, all four concentrations were effective in increasing fibrosis-related genes, and 5 ng/ml was selected. In addition, this paper observed that the protein expression levels of CTGF, Fibronectin and α-SMA were induced in the orbital fibroblasts by treatment of 5 ng/ml TFGβ1 for 12, 24, 48, and 72 hours, respectively. As a result, the induction of fibrosis-related proteins by 5 ng/ml TGF-β1 treatment for 24 hours was most effective in orbital fibroblasts from GO patients.

  1. If add TGFβ directly to the co-culture system of hPMSCs/ TAO OFs, do they have a similar effect?

-> As you mentioned, we were also curious about the direct effect of TGFβ on orbital fibroblasts. Therefore, we conducted the experiment in order to evaluate the expression of fibrotic genes in TAO-OFs as shown above.  When TAO-OF was treated with only TGFβ (20ng/mL), fibrosis markers such as α-SMA and Fibronectin levels increased. Only when TGFβ (20ng/mL) treated hPMSCs were co-cultured with TAO-OFs, anti-fibrotic effect was achieved as well as anti-lipogenesis and anti-adipogenesis.

  1. There are many English typos.

-> I am very afraid that there were many typos. I corrected them all.

I hope that the revised one would be Ok with you.

Reviewer 2 Report

The manuscript is interesting. There are however the following points that should be taken into account:

  1. The authors describe that cell viability of hPMSCs decreased by 25.9% (line 58). However, they measure MTT, which allow to determine the activity of viable cells (not the ones that are dead and do not maintain the pH differences in the mitochondria). However, the assay does not allow to determine if a fraction of the cells are dead or not. Practically speaking the authors cannot distinguish between 25.9% of dead cells in the TGFbeta treated cultures or that all the cells in this condition are alive but 25.9% of the ones untreated proliferated and therefore there is an increase in this population of 25.9% of MTT activity. The authors should clarify the findings and if possible should look by other methods which of the two possibilities is the correct one and describe it.
  2. On line 145 the phrase starting by: To investigate...... is not clear, please re-write it making clear what do they mean.
  3. Similarly, on lines 213 and 214, the phrase starting by: The role of TGFbeta in inflammatory disease is defective TGFbeta..... is not clear. Clarify, please, it looks to me like a copy/paste from another place and does not make sense
  4. In materials and methods (point 4.7, lines 305 and 306) where it says ...A subunit plasmid to leg muscle.... please correct, as far as I am aware subunit plasmids do not exist.
  5. In lines 307 and subsequent lines, the authors describe that the animals ... were divided into four groups and injected with .... please state the injection site (intramuscular, intraperitoneal, i.v., etc) it may well be infraorbital but it is not clear, please clarify.
  6. in line 342, the informed consent statement it says not applicable, but on lines 250-251 it states that all patients' consent was obtained, please clarify the incongruence.
  7. In figure 1C the data for TNFalpha shows an increase, but the authors do not show any significance of the increase (I believe they forgot to label it, please correct). 
  8. on line 62 it states that mRNA expression of lipogenesis marker genes IGF-1.... were increased, however the figure 1B does not show any increase for this particular gene, please clarify.
  9. In some figures, (see for example figure 2) there is a lot of blank space and the panels are too small and really difficult to read and follow, please enlarge the panels leaving les blank space and this would give a better view of the image

Minor points:

  1. line 61, change wre for were 
  2. line 62, change expressions (in mRNA expressions) for expression
  3. line 89, where it says: was more effective than change for: more effectively than
  4. line 95: is involved in other reaction change for: was involved in other reactions
  5. line 98 where it says: more so than... change for: more effectively than...
  6. line 110, where it says: qRT-PCR change for: RT-qPCR (the quantitative reaction is the PCR not the RT)
  7. throughout the text, where it says means ± SEM, change for mean± SEM (starting on line 127, but on several places, please check)
  8. line 197, where it says interacts change for interact
  9. line 210: change that line for: habits adipogenesis, this inhibition is rescued by the addition of the SB431542 inhibitor of SMAD2/3 [22-24]
  10. line 239: change: researches for research
  11. line 244: change gene for genes
  12. line 258 change using fifth to eighth cell passage for: using cells from fifth to eighth passage.
  13. line 269: change Druing for During
  14. line 275 change: was treated recombinant change for: wee treated with recombinant
  15. line 297, the phrase starting with According to... change for: According to the manufacturer's protocol,
  16. line 322: where it says decomposed into change for: separated by

Author Response

Answers to the reviewers’ comments

I read all the comments and rewrote the phrases according to your suggestions.

And I also tried to explain the ideas and answered to every question from you.

I really appreciate your professional comments and excellent points to improve my manuscript for the readers.

Reviewer 2

Major points:

  1. The authors describe that cell viability of hPMSCs decreased by 25.9% (line 58). However, they measure MTT, which allow to determine the activity of viable cells (not the ones that are dead and do not maintain the pH differences in the mitochondria). However, the assay does not allow to determine if a fraction of the cells are dead or not. Practically speaking the authors cannot distinguish between 25.9% of dead cells in the TGFbeta treated cultures or that all the cells in this condition are alive but 25.9% of the ones untreated proliferated and therefore there is an increase in this population of 25.9% of MTT activity. The authors should clarify the findings and if possible should look by other methods which of the two possibilities is the correct one and describe it.

-> We really agree with your comments about the explanation of viable cell assay test including MTT activity. And we ask you apologies for the wrong description of methods. The test previously performed was not MTT but CCK-8 test. Therefore, we rewrote about the result of CCK-8 in Fig.1A.

In addition, we also performed BrdU assay (Cat#CBA-251, Cell Biolabs, USA) to measure cell proliferation activity, as you recommended. When treated TGFβ (20ng/mL) to hPMSCs and incubate for 24h, cell proliferation activity significantly decreased by 39.2% in TGFβ treated hPMSCs. The graph shown above was added in Fig. 1B.

In conclusion, it was confirmed that TGFβ-hPMSCs presented low level of CCK-8 activity and proliferation activity, but they also demonstrated the decreased expression of lipogenic and inflammatory markers IL6, ICAM and IL-1β and increased expression of anti-lipogenic gene such as INSIG2. Furthermore, in the results of the in vivo model, it was confirmed that TGFβ-hPMSCs played a positive role by anti-adipogenesis.

  1. On line 145 the phrase starting by: To investigate...... is not clear, please re-write it making clear what do they mean.

-> We rewrote the phrase to make it clear.

: To investigate the associated pathway regarding inhibition of adipogenesis and lipogenesis, and this content was added to the result section 2.5.

  1. Similarly, on lines 213 and 214, the phrase starting by: The role of TGFbeta in inflammatory disease is defective TGFbeta..... is not clear. Clarify, please, it looks to me like a copy/paste from another place and does not make sense

-> We rewrote the phrase to make it clear.

: Sana, S. et al. proposed the relationship between SMAD proteins and TGFβ signaling, which high SMAD7 immunoreactivity and lack of p-SMAD3 expression caused a defective TGFβ signaling in environmental enteropathy [27] and it was similar to the previous result in inflammatory bowel disease and Crohn’s disease [28], and this was added content to the discussion section.

  1. In materials and methods (point 4.7, lines 305 and 306) where it says ...A subunit plasmid to leg muscle.... please correct, as far as I am aware subunit plasmids do not exist.

We rewrote the phrase to make it clear.

: We were provided TSHR-A subunit plasmid from Prof. Banga in UK. He has been actively researching GO models. In addition to the papers we reported, many papers on the characteristics of the GO model have been published (PMID: 21715431, 23900776, 26872090, 34051850, 31399042). I added this content to the method section 4.7.

New reference with number of 36 was added.

(Zhao, S. X.; Tsui, S.; Cheung, A.; Douglas, R. S.; Smith, T. J.; Banga, J. P., Orbital fibrosis in a mouse model of Graves' disease induced by genetic immunization of thyrotropin receptor cDNA. The Journal of endocrinology 2011, 210, (3), 369-77.)

  1. In lines 307 and subsequent lines, the authors describe that the animals ... were divided into four groups and injected with .... please state the injection site (intramuscular, intraperitoneal, i.v., etc) it may well be infraorbital but it is not clear, please clarify.

-> We rewrote the phrase to make it clear.

: We injected the mouse in to the orbit (intra orbital injection) and added this content to the method section 4.7.

  1. in line 342, the informed consent statement it says not applicable, but on lines 250-251 it states that all patients' consent was obtained, please clarify the incongruence.

-> We rewrote the phrase to make it clear.

: This research was approved by the Institutional Review Board of Bundang CHA Medical Center, Seongnam, Republic of Korea (IRB-2018-01-007). I modified the Institutional Review Board Statement part by adding the IRB number.

  1. In figure 1C the data for TNFalpha shows an increase, but the authors do not show any significance of the increase (I believe they forgot to label it, please correct).

-> We rewrote the phrase to make it clear.

: In Figure 1C, TGFβ2 asterisk was modified and asterisk was displayed on the TNFα graph.

  1. on line 62 it states that mRNA expression of lipogenesis marker genes IGF-1.... were increased, however the figure 1B does not show any increase for this particular gene, please clarify.

-> We rewrote the phrase to make it clear.

: I deleted IGF-1 word in results section 2.1, so please review it.

  1. In some figures, (see for example figure 2) there is a lot of blank space and the panels are too small and really difficult to read and follow, please enlarge the panels leaving les blank space and this would give a better view of the image

-> We modified the panels to make it stand out with a better view.

Minor points:

->We revised all points mentioned by the reviewer to the manuscript and painted the modified part in yellow in the revised one.

  1. line 61, change wre for were

We rewrote the word to correct the typo.

  1. line 62, change expressions (in mRNA expressions) for expression

We rewrote the word to correct the typo.

  1. line 89, where it says: was more effective than change for: more effectively than

We rewrote the word to correct the typo.

  1. line 95: is involved in other reaction change for: was involved in other reactions

We rewrote the word to correct the typo.

  1. line 98 where it says: more so than... change for: more effectively than...

We rewrote the word to correct the typo.

  1. line 110, where it says: qRT-PCR change for: RT-qPCR (the quantitative reaction is the PCR not the RT)

We rewrote the word to RT-qPCR.

  1. throughout the text, where it says means ± SEM, change for mean± SEM (starting on line 127, but on several places, please check)

We rewrote the word to correct the typo (means to mean) in the legends of figure 2, 3 and 4.

  1. line 197, where it says interacts change for interact

We rewrote the word to correct the typo.

  1. line 210: change that line for: habits adipogenesis, this inhibition is rescued by the addition of the SB431542 inhibitor of SMAD2/3 [22-24]

We rewrote the word to correct the typo.

  1. line 239: change: researches for research

We rewrote the word to correct the typo.

  1. line 244: change gene for genes

We rewrote the word to correct the typo.

  1. line 258 change using fifth to eighth cell passage for: using cells from fifth to eighth passage.

We rewrote the word to correct the typo.

  1. line 269: change Druing for During

We rewrote the word to correct the typo.

  1. line 275 change: was treated recombinant change for: wee treated with recombinant

We rewrote the word to correct the typo.

  1. line 297, the phrase starting with According to... change for: According to the manufacturer's protocol,

We rewrote the word to correct the typo.

  1. line 322: where it says decomposed into change for: separated by

We rewrote the word to correct the typo.

Round 2

Reviewer 2 Report

none